# One-Pot Synthesis of Pt High Index Facets Catalysts for Electrocatalytic Oxidation of Ethanol

**DOI:** 10.3390/nano12244451

**Published:** 2022-12-14

**Authors:** Ruihua Guo, Na An, Yarong Huang, Lili Guan, Guofang Zhang, Guofu Zhu, Zhaogang Liu

**Affiliations:** 1College of Materials and Metallurgy, Inner Mongolia University of Science and Technology, Baotou 014010, China; 2Inner Mongolia Key Laboratory of Advanced Ceramics and Device, Inner Mongolia University of Science and Technology, Baotou 014010, China; 3Key Laboratory of Green Extraction and Efficient Utilization of Light Rare Earth Resources of Ministry of Education, Inner Mongolia University of Science and Technology, Baotou 014010, China

**Keywords:** high index facets, electrocatalyst, electrocatalytic oxidation, direct ethanol fuel cell

## Abstract

Direct ethanol fuel cell (DEFC) has attracted wide attention due to its wide range of fuel sources, cleanliness, and high efficiency. However, the problems of low catalytic efficiency and poor catalyst stability still exist in DEFC catalysts, which restrict its rapid development. With chloroplatinic acid (H_2_PtCl_6_·6H_2_O) as the precursor, Polyvinylpyrrolidone (PVP) plays the role of surfactant, stabilizer, and reducing agent in the experiment. Glycine is the surface control agent and co-reducing agent. Pt high-index facets nanocatalyst was prepared with the one-pot hydrothermal method by adjusting the amount of PVP and glycine. X-Ray Diffraction (XRD), transmission electron microscope (TEM), and scanning electron microscope (SEM) were used to characterize the micro-structure of the nanocatalyst, and the influence of PVP and glycine on the synthesis of high-index facets catalyst was studied. The electrocatalytic performance of the catalyst was tested with an electrochemical workstation, and it was found that the performance of the prepared catalyst was better than that of the commercial catalyst. When the mass ratio of PVP and Pt was 50:1 and the molar ratio of glycine and Pt was 24:1, Pt nanocatalysts with {310}, {520} and {830} high exponential facets were prepared. The electrochemical test results showed that the peak current density of ethanol oxidation was 2.194 m^2^/g, and the steady-state current density was 0.241 mA/cm^2^, which was 5.7 times higher than that of commercial catalyst. The results of this paper show that due to the defects such as steps and kinks on the surface of the high-index facets, the active sites are increased, thus showing excellent electrocatalytic performance. This study provides a theoretical basis for the development and commercial application of high index facets nanocatalysts.

## 1. Introduction

With the gradual depletion of fossil fuels and the deterioration of human living environments, traditional power generation methods, such as thermal power generation, wind power generation, and water conservancy power generation, are facing a series of serious challenges due to geographical problems, pollutant emissions, the applicability of the surrounding environment, and power loss. Fuel cells, as an independent power generation device, can be used for centralized power generation and decentralized power generation. It not only has strong applicability to the surrounding environment and low power loss, but also is not affected by the Carnot cycle in the power generation process [1,2]. Therefore, it has attracted the attention and research of many countries. Direct Ethanol Fuel Cell (DEFC) is a new kind of fuel cell which is very popular in recent years. As the simplest organic small molecule, ethanol has the advantages of high theoretical energy density (8.1 kW h/kg), low permeability, safety, non-toxic, and wide source. The theoretical energy conversion efficiency of complete combustion is as high as 0.969, which is more than that of a methanol fuel cell (0.967) and hydrogen oxygen fuel cell (0.83) [3]. Therefore, it can be used as a new energy in various fields. However, the conversion efficiency of a DEFC oxidation reaction is low in commercial application. The reasons for this phenomenon include the chain structure of ethanol, catalyst type, fuel structure, membrane used and other physical factors (concentration, temperature, etc.). Only by breaking through the expensive cost and inferior efficiency of the catalysts, improving the electrocatalytic efficiency of the catalyst, accelerating the oxidation of ethanol molecules, and improving the efficiency of the battery can DEFC be commercially produced as soon as possible.

At present, the DEFC electrocatalyst mainly uses Pt as the main active component, and other transition metals and oxides are added as the auxiliary catalysts. On this basis, a certain amount of carrier is loaded as the dispersant, binder, and support of the active component. In addition, the appropriate microstructure was synthesized by adding some surfactants and controlling agents or adjusting the synthesis process parameters to reduce the adsorption number of poisons on the Pt surface and accelerate the C-C bond cleavage of ethanol, so that it can be completely oxidized to improve the energy conversion rate of ethanol [4,5,6]. Considerable basic studies on Pt single crystal surface have shown that the uncovering high-index facets of catalyst nanoparticles usually have more preferable catalytic activity than the most common low-index facets (such as {111}, {100} and {110}). This is because there are a large number of high-density atomic steps, protrusions and kinks on the high-index facets, which increase the active sites involved in the catalytic reaction and destroy the chemical bonds [7]. For example, Pt {210} crystal facet has high catalytic activity for the electroreduction of CO_2_ and the electrocatalytic oxidation of formic acid [8]. Pt {410} crystal facet shows high efficiency for the catalytic and decomposition of NO, which is a major pollutant in automobile exhaust. Therefore, the morphology controllable synthesis of high index facets is a potential way to improve the catalytic activity of nanocrystals. However, due to the high surface energy of the high-index facets, it is very arduous to synthesize the nanocrystals surrounded by the high-index facets.

In 2007, Sun et al. [9] notified the exposure of tetrahexahedral Pt single crystals with 24 high index facets, such as {730}, {210} and {520}. Since these high index facets have large atomic step density and twist, they can obviously enhance the electrocatalytic activity of the catalyst. Since then, people’s understanding has changed from shape-controlled synthesis to reasonable construction of high-index facets. Therefore, in the research of microstructure of metal oxides, the concept of high-index facets has also received considerable attention. Especially in recent years, the study of high-index facets structure has increased significantly. By revealing the relationship between high-index facets and catalytic activity, highly efficient catalysts with high-index facets were prepared. However, in the process of synthesizing high-index facets by electrochemical method, the synthesized high-index facets are relatively single, and cannot form quite a lot of high-index facets. Therefore, it is urgent to find a method to synthesize quite a lot of catalysts with high-index facets once. Zheng et al. [10] notified the preparation of concave polyhedron platinum nanocrystals with {411} facets with the hydrothermal method. It was synthesized by the reduction of hexachloroplatinic acid with amine and polyvinylpyrrolidone (PVP) in N, N-dimethylformamide (DMF). During the growth of Pt nanocrystals, the exposed {411} planes are retained due to the coordination of amines, which helps to holding the low coordination Pt sites. Zhang et al. [11] effectively generated Pt concave nanotubes with different angles on GO by one-pot hydrothermal method. With H_2_O_2_ as the research object, the electrocatalytic behaviors under different conditions were analyzed. Electrochemical experiments show that the maximum retention current of high-index facets is at least 4 times higher than that of low-index facets. Wang et al. [12] synthesized tetrahedral platinum nanocrystals surrounded by 24 {730} high-index facets with the seed technique using silver ions as seeds and cetyltrimethylammonium bromide (CTAB) as a morphology control agent. Mirkin et al. [13] prepared a new class of gold nanostructures in a monodisperse manner using an improved seed-mediated method: namely, concave nanocubes wrapped in 24 high-index {720} facets. Compared with low-index {111} octahedrons, concave nanocubes showed higher electrochemical activity. Lee et al. [14] synthesized uniform and controllable concave trioctahedral gold nanocrystals with the seed-mediated method using cetyltrimethylammonium chloride (CTAC) as a capping agent. The analysis of atomic arrangement and projection angle showed that the surface of concave nanotubes had {221}, {331}, and {441} high refractive index {hhl} crystal facets. Han et al. [15] prepared Au nanocrystals with hexahedral structure surrounded with {321} facets. Under the action of appropriate surfactant, controlling the growth kinetics of NC by controlling the addition of reducing agent and reaction impedance temperature is the key synthesis method to control the morphology of Au NCs. It is precisely because of its unique morphological characteristics that hexahedral gold nanotubes exhibit excellent optical and surface enhanced Raman scattering activity. Xia et al. [16], as an early researcher to explore high-index facets catalysts, synthesized different high-index Pt nanocrystals by adjusting the reductant and stabilizer. By controlling the proportion rate of NaBH_4_, Pt-based catalysts with high-index facets with uniform nanoparticles distribution and high catalytic performance were synthesized at room temperature. It was found that the crystal planes exposed on the synthesized catalyst surface were {720}, {510} and {830} crystal planes. Xia et al. synthesized concave cubes coated with {730} facets using Na_2_PdCl_4_ as Pd source, ascorbic acid as a reducing agent, and Br^-^ as etching agent. The results showed that the catalytic activity of high-index facets {730} was doubled compared with that of low-index facets. Therefore, the study of catalysts with high index facets is very necessary for the promotion and application of fuel cells.

In this paper, high index Pt nanocatalysts with high index facets were successfully synthesized in aqueous solution by reducing H_2_PtCl_6_·6H_2_O with PVP k-30 and glycine in aqueous solution. The one-pot method has the advantages of simple process flow, short reaction cycle and suitable for industrial production. A series of Pt high-index facets catalysts were prepared by adjusting the amount of PVP and glycine. The catalysts were characterized by XRD, SEM and TEM. The electrochemical activity and stability of Pt concave nanocubes with high-index crystal surface were studied with electrochemical tests.

## 2. Experimental

### 2.1. Materials

Hydrochloroplatinic acid (H_2_PtCl_6_·6H_2_O, ≥37%(Pt)) was obtained from Tianjin Comiou Chemical Reagent Co., Ltd. (Comiou, Tianjin, Chiina). Polyvinylpyrrolidone K30 (PVP) ((C_6_H_9_NO)_n_, analytical purity) and Pt/C (JM) were obtained from China National Pharmaceutical Group Co., Ltd. Glycine (C_2_H_5_NO_2_, analytical purity) was purchased from Macklin Biochemical Co., Ltd. (Macklin, Shnghai, China). Nafion (5%) solution (analytical purity) was obtained from Alfa Aesar. Concentrated sulfuric acid (H_2_SO_4_, ≥98%) and anhydrous ethanol (C_2_H_5_OH, ≥99.7%) were purchased from Fengchuan Chemical Reagent Co., Ltd. (Fengchuan, Tianjin, Chiina). The water used in the experiments was ultrapure water.

### 2.2. Synthesis of Catalyst with PVP as Variable

PVP (K30) with different mass ratios (mass ratio: PVP:Pt = 0:1, PVP:Pt = 25:1, PVP:Pt = 50:1, PVP:Pt = 75:1, PVP:Pt = 100:1) and 675 mg glycine were added to 63 mL deionized water for ultrasonic dispersion for 30 min and magnetic stirring for 30 min. Then 9.32 mL aqueous solution of chloroplatinic acid was added to the above dispersed mixed solution while stirring. After complete mixing, the above mixed solution was shifted to a 100 mL PTFE reactor. The reaction was heated at 200 °C for 7 h, and the reaction was taken out and natural cooling. The cooled solution was cleaned with ultrapure water and ethanol and dried at 60° for 5 h.

### 2.3. Synthesis of Catalyst with Glycine as Variable

Glycine with different molar ratios (molar ratio: Gly:Pt = 0:1, Gly:Pt = 8:1, Gly:Pt = 16:1, Gly:Pt = 24:1, Gly:Pt = 32:1) and 3.6 g PVP (K30) were added to 63 mL deionized water for ultrasonic dispersion for 30 min and magnetic stirring for 30 min. Then, 9.32 mL aqueous solution of chloroplatinic acid was added to the above dispersed mixed solution while stirring. After complete mixing, the above mixed solution was shifted to a 100 mL PTFE reactor. The reaction was heated at 200 °C for 7 h, and the reaction was taken out and natural cooling. The cooled solution was cleaned with ultrapure water and ethanol and dried at 60° for 5 h.

### 2.4. Preparation of Working Electrode

The test used in this pattern is a three-electrode test system. The working electrode is a glassy carbon electrode, the platinum wire is a counter electrode, and the saturated calomel electrode is used as a reference. The specific preparation process of the working electrode is as follows: first, the glassy carbon electrode was polished to the mirror surface with 0.3 μm Al_2_O_3_ powder on the polishing cloth. After polishing, the glassy carbon electrode was washed three times with water and ethanol in ultrasonic wave and then dried quickly. After connecting the electrochemical test workstation, in 0.5 mol/L H_2_SO_4_ scan, test whether appear stable curve. Then, the cyclic voltammograms were tested in 0.005 mol/L K_3_[Fe(CN_6_)] and 0.1 mol/L KCl, and the peak spacing was required not to exceed 80 mV. At this time, the catalyst solution was dropped on the working electrode. First, a mixed solution containing 5 mL deionized water and 5 mL anhydrous ethanol was prepared. Then 10 mg catalyst powder was mixed with the above solution to form a slurry, which was ultrasonically treated for 30 min to fully disperse. A microsampler was used to transfer 6 μL of liquid to the glassy carbon electrode, and 2 μL of Nafion solution was transferred to the glassy carbon electrode after drying. After natural drying for the next experiment.

### 2.5. Characterization

The composition and crystallization of the catalysts were analyzed by Miniflex600 X-ray diffractometer (XRD, Rigaku, Shanghai, China). The working voltage was 40 kV, the working current was 15 mA, the light source was Cu Kα ray source, the wavelength was 0.15406 nm, the scanning speed was 2°/min, and the scanning range was 10°~90°. Thermal field emission scanning electron microscope Apollo300 (SEM, SAZXIT, Shanghai, China) was used to scan the catalysts. The morphology, particle size and distribution of the synthesized catalysts were analyzed by JEM-2100 transmission electron microscope (TEM, JEOL, Tokyo, Japan).

### 2.6. Electrochemical Characterization

The electrochemical test of the synthesized catalysts was carried out using the IVIUM electrochemical workstation produced in the Netherlands. The test used a three-electrode system, a platinum wire counter electrode, a saturated calomel electrode as a reference electrode, and the potential was relative to the saturated calomel electrode. The electrochemical active surface area (ESA) was measured in 0.5 mol/L H_2_SO_4_ solution, the scanning potential range was −0.3–0.6 V, and the scanning speed was 50 mV/s. Cyclic voltammetry (CV) electrochemical test using 1 mol/L CH_3_CH_2_OH + 0.5 mol/L H_2_SO_4_ solution, scanning potential range of 0.0V~1.2V, and scanning speed of 50 mV/s. A chronoamperometry test using a mixed solution of 1 mol/L CH_3_CH_2_OH + 0.5 mol/L H_2_SO_4_, polarization potential of 0.6 V, and set test time of 1100 s. The mixed solution of 1 mol/L CH_3_CH_2_OH + 0.5 mol/L H_2_SO_4_ was used in the variable temperature cyclic voltammetry test. The scanning potential range was 0.0V~1.2V, and the scanning speed was 50 mV/s. The temperature was set at 25 °C, 30 °C, 35 °C, 40 °C, 45 °C, 50 °C, 55 °C and 60 °C. The number of test cycles was set to 10 cycles. Depletion cyclic voltammetry test electrolyte, potential test interval, test rate and temperature cyclic voltammetry test, set the number of cycles of cyclic voltammetry test failure 500 laps. Before all electrochemical tests, N_2_ was introduced into the solution for about 30 min to remove the effect of O_2_.

## 3. Results and Discussion of Catalysts with PVP as Variable

### 3.1. Characterization

#### 3.1.1. X-ray Diffraction (XRD) Characterization of the Catalysts

In order to characterize the crystal structure of the synthetic catalysts, XRD tests were performed on the catalysts prepared by adjusting different PVP contents, as shown in Figure 1. It can be found from the figure that the prepared catalysts all had five peaks in the range of 10°–90°. Compared with the Pt standard PDF card, it was found that the peaks at 39.7° corresponded to Pt (111) crystal plane, 46.2° corresponded to Pt (200) crystal plane, 67.5° corresponded to Pt (220) crystal plane, 81.3° corresponded to Pt (311) crystal plane, 85.7° corresponded to Pt (222) crystal plane, indicating that Pt^4+^ in the prepared catalysts was successfully reduced to Pt.

#### 3.1.2. SEM Characterization of the Catalysts

It was found that the morphology of the nanoparticles was also affected by the surfactant PVP. For the purpose of analyze the effect of PVP on the morphology of Pt high exponential facets (Pt HIFs) nanocatalysts, five control groups with different PVP additions were prepared. The mass ratios of PVP to Pt were PVP:Pt = 0:1, PVP:Pt = 25:1, PVP:Pt = 50:1, PVP:Pt = 75:1, PVP:Pt = 100:1, respectively. The catalysts were characterized by SEM, as shown in Figure 2. When the surfactant PVP was not added in the reaction process, the obtained nanoparticles were all irregular polyhedrons the particle size was greater than 30 nm, resulting in serious agglomeration of catalyst nanoparticles. When PVP is added, the dispersion becomes better and better with the increase of PVP. When PVP:Pt = 25:1, the catalyst nanoparticles are some large irregular particles and concave cubes. With the addition of PVP increased to PVP:Pt = 50:1, the catalyst nanoparticles showed concave cubic morphology. When the amount of PVP was increased to PVP:Pt= 75:1, the nano-particles still maintained the concave cube morphology, indicating that the morphology of the catalyst was stable, but the particle size was large at this time. When the amount of PVP was further increased to PVP:Pt = 100:1, the catalyst was still concave at this time, but the concave degree was small, and the particle size was large. The experiments show that PVP not only acts as a reducing agent in the reaction process, but also acts as a protective agent and dispersant to protect the dispersion of nanoparticles. When the amount of PVP is large, the growth rate of nanoparticles is accelerated, resulting in the growth of nanoparticles [17].

#### 3.1.3. TEM Characterization of the Catalysts

For the purpose of further observe the morphology of Pt high-index facets nanocatalyst, Figure 3 was obtained by TEM and HTEM. It can be obtained from Figure 3a,b that Pt nanocrystals show uniform concave cube morphology and uniform dispersion, and most of the catalyst nanoparticles form concave cube morphology. The concave cube is a typical morphology with a high-index facets {hk0}. Through TEM images, it can be observed that each face of the prepared nanoparticles is concave, and each concave is composed of four faces (as shown in the upper right corner of Figure 3a). At the same time, the particle size distribution of the concave cube nanoparticles is counted, and the average size of the measured particles is about 26.72 nm. M.A. Van Hove [18] found that the high-index facets can be expressed by calibrating the high-index facets from the defects of step, twist and step, and the exposed and high-index facets information of the catalyst can be determined by measuring the concave angle of the catalyst nanoparticles [19,20]. Therefore, the concave angle of the particles with concave cube morphology was measured with the above method, and the crystal surface information of the prepared catalyst was calibrated as shown in Figure 3c. The concave angles of the concave cube block are measured to be 17.3°, 19.6°, 19.1°, 20.6°, 23.3°, 22.5°, 18.5° and 19.5° (clockwise). The high-index facets corresponding to each angle are listed in Table 1.

According to the high index crystal plane index corresponding to these angles, it is found that the main exposed high index crystal planes of Pt high index crystal nanocatalysts are {310}, {520} and {830}. Figure 3e–g are the corresponding model diagrams of these three crystal planes. It indicated that the nanocatalyst with high index facets could be successfully synthesized under the appropriate PVP and glycine addition.

In the synthesis process, it was found that PVP mainly acted as a reducing agent and a dispersant in the synthesis of high-index facets, and it played a very necessary role in the synthesis of high-index concave Pt nanoparticle catalysts [21]. By adjusting the addition amount of PVP, the influence of PVP on high-index facets was discussed. It can be observed from Figure 4a,b that when the mass ratio of PVP the mass ratio of PVP to Pt was PVP:Pt = 0:1, the synthesized catalyst nanoparticles showed very serious agglomeration, and the nanoparticles could not be completely dispersed. At this time, most of the synthesized nanoparticles were irregular polyhedrons. The average size of Pt-based catalysts was found to be 32.08 nm by particle size statistics. The concave angle of the clear nanoparticles was measured, and it was found that the exposed crystal faces were mainly {710} and {610}. 

It can be seen from Figure 4c,d that when the mass ratio of PVP to Pt was PVP:Pt = 25:1; it can be observed that the dispersion of the synthesized catalyst nanoparticles was greatly improved compared with that without PVP. However, it can be observed that the nanoparticles still exist agglomeration phenomenon. At this time, it is found that the nanoparticles begin to appear some concave cubes, but there are still a large number of irregular polyhedrons. At the same time, the particle size statistics of nanoparticles show that the average size of Pt nanoparticles is 28.67 nm. The concave angle measurement shows that the crystal faces exposed to the base are mainly {610}, {520} and {410}. When the mass ratio of PVP to Pt was PVP:Pt = 50:1, it can be observed from Figure 4e,f that there was almost no agglomeration of the synthesized catalyst nanoparticles. At this time, the dispersion of the catalyst nanoparticles was uniform, which was greatly improved compared with the previous two groups. At this time, it was observed that the synthesized nanoparticles had a large number of concave cube morphologies, and there was almost no irregular polyhedron.

The particle size statistics of nanoparticles showed that the average size of Pt-based catalysts was 26.72 nm. At the same time, the nanocatalyst with concave surface was selected to measure the concave angle. At this time, the exposed high-index crystal planes were mainly {310}, {520} and {830}. The concave degree further increased, the corresponding step atomic density increased, and the active sites increased [22]. When the mass ratio of PVP to Pt is PVP:Pt = 75:1 and PVP:Pt = 100:1, it is found from Figure 4g–j that the synthesized catalyst nanoparticles are no longer agglomerated, and the dispersion of nanoparticles is very uniform. The nanoparticle size statistics of the two groups of catalysts are carried out, and it is found that the nanoparticle sizes of the two groups of catalysts are 27.07 nm and 31.2 nm, respectively. The concave angle measurement shows that the high index facets exposed at this time are mainly {410} and {720}. It was also found that with the increase of PVP content, the concave surface of the synthesized catalyst became smaller and smaller, and the grain size of the catalyst became larger and larger.

### 3.2. Electrochemical Characterization

#### 3.2.1. Electrochemical Active Surface Area of Catalyst

With the purpose of investigate the catalytic performance of the prepared catalyst, the electrocatalytic performance of the catalyst was investigated. Figure 5 shows the electrochemical characteristic test of the prepared samples and commercial sample (Pt/C(JM)). When the catalytic activity is measured, the electrochemical active surface area (ESA) of the catalyst is usually used to represent the calculation process. The double-layer capacity is excluded. The adsorption-desorption curve of the catalyst for hydrogen is obtained with testing, and the scanning curve is integrated [23,24]. The electrochemical active area of the catalyst can be calculated with the following formula [25]:ESA = Q_H_/(0.21 × [Pt])(1)
Q_H_ = S/v(2)

In the formula, ESA represents the electrochemical active area; Q_H_ represents the amount of electricity adsorbed or desorbed by H; [Pt] represents the mass of Pt adsorbed on the electrode surface; S represents the area of adsorption or desorption peaks obtained by integration, which can be obtained by integrating the peaks; V Represents a scanning speed of 50 mV/s; 0.21 mC/cm^2^ represents the amount of electricity required for Pt to adsorb H per unit area.

It can be seen from Figure 5 (local magnification at the lower right corner) that the oxidation peak appeared in the range of −0.3V~−0.1V during the positive scanning process, and the reduction peak appeared in the range of −0.2V~−0.3V during the negative scanning process. According to the formula 1 and2, the electrochemical active area (ESA) in Table 2 can be calculated. It can be found from the calculation results that compared with commercial Pt/C(JM) catalysts, the electrochemical active surface area of the prepared high-index crystal surface catalysts has been greatly improved, and the addition amount of PVP in the reaction process is adjusted. With the increase of PVP content, the electrochemical active surface area increases first and then decreases. When the addition of PVP is PVP:Pt = 0:1, the electrochemical active area of the catalyst is very low, which is due to the serious agglomeration of the catalyst without PVP and the insufficient exposure of the active site, resulting in low electrochemical active area and low electrochemical performance. When the PVP content increased to PVP:Pt = 50:1, the electrochemical active area reached the maximum value of 2.194 m^2^/g. It was found that PVP, as a dispersant and auxiliary reducing agent, played an essential role in the synthesis of samples in the synthesis of high index facets. When PVP is not added in the synthesis system, the synthesized catalyst grains will appear serious agglomeration phenomenon and the dispersion is very low, which will cause that the active sites that can participate in the catalytic oxidation of ethanol are not fully exposed, so the catalytic oxidation activity decreases and the electrochemical active area decreases.

#### 3.2.2. Cyclic Voltammetry of the Catalysts in Ethanol Oxidation

In an effort to further explore the catalytic oxidation performance of the prepared catalyst for ethanol, the cv curves in 0.5 mol/L H_2_SO_4_ + 1 mol/L CH_3_CH_2_OH solution were tested. As can be seen from Figure 6a, there are three peaks in the cyclic voltammetry curve during the scanning process. In the forward scanning process, ethanol is completely oxidized to CO_2_ in the range of 0.4 V~0.8 V, and the peak in this range is usually used as the standard to evaluate the catalytic performance of the samples. With the increase of voltage, the oxidation peak begins to decrease. This is because ethanol will produce some poisons adsorbed on the appearance of Pt catalyst in the oxidation process, which inhibits the catalytic activity of Pt and reduces the current density [26]. Then, in the range of 0.8 V~1.1 V during the positive scanning process, the peak appeared as incomplete oxidation of ethanol to acetaldehyde. At the same time, Pt was oxidized to PtO due to the adsorption of −OH, inducing in the decrease of catalyst activity. Current density decreases as Formula (3) [27,28,29]:Pt-OH → PtO + H^+^ + e^−^(3)

But when PtO is reduced to Pt during negative sweep, the catalyst restores its catalytic activity, such as Formulas (4) and (5).
PtO + H_2_O + e^−^ → Pt-OH + H^+^(4)
PtO + H^+^ + e^−^ → Pt + H_2_O(5)

Because the first peak in the positive scan is the peak of complete oxidation of C_2_H_5_OH to CO_2_, which is used as a standard to measure the catalytic performance of the samples [30].

It can be found in Figure 6a that the peak current density of the prepared high-index facets catalyst has been greatly improved compared with that of the commercial Pt/C(JM) catalyst. This is because there are a large number of step atoms and twist atoms on the appearance of the high-index facets, which can be used as the active sites for catalytic ethanol. It is easier to contact with ethanol and interrupt the chemical bond of ethanol to become the center of the chemical reaction. The nanoparticles are in a compressive strain state near the surface, a decrease in the distance between atoms shifts the d-band center to a higher energy. It can be seen from the figure that with the enhancement of PVP content, the oxidation current density of the catalyst for ethanol first added and then receded. When PVP:Pt = 50:1, the maximum oxidation current density for ethanol reached 2.052 mA/cm^2^, about 5.8 times that of commercial Pt/C catalyst, suggesting that the catalyst synthesized under this condition had the highest catalytic activity for ethanol. This is because when PVP is not added, the formed high index facets nanocrystals will agglomerate to form some large particles, and the active sites cannot be fully exposed to reduce the oxidation current density. With the increase of PVP content, the high-index facets grain began to disperse gradually, and they reached the best when PVP:Pt = 50:1. When the amount of PVP continues to increase, PVP as a reducing agent, the catalyst grain continues to grow, the specific surface area decreases, resulting in the oxidation of ethanol current density began to decrease, which is corresponding to the analysis results in Figure 6b.

#### 3.2.3. I–t Curves of the Catalysts

In an effort to further investigate the stability of the prepared catalyst, the chronoamperometry was carried out in 1 mol/L 0.5 mol/L H_2_SO_4_+CH_3_CH_2_OHsolution. As shown in Figure 7, it can be found that the current density decreases gradually with the increase of time and tends to be stable at about 1000 s. The reason for this phenomenon is that at the beginning of the test, the ethanol concentration near the electrode is high, the thickness of the diffusion coating is small, and the current density will be very high. However, with the continuation of the reaction, the ethanol molecular concentration of the electrode adjuvant is decreasing, and the incomplete oxidation products of ethanol will also accumulate near the electrode, such as CO, which will be adsorbed on the surface of the active site of the Pt catalyst, inducing the catalyst cannot continue to catalyze the oxidation of C_2_H_5_OH, and the current density is getting lower and lower.

Therefore, the timing current test of the catalyst can measure the stability and anti-poisoning performance of the catalyst [31]. As shown in Table 2, when PVP:Pt = 50:1, the oxidation current density of the catalyst in the chronoamperometry test after 1100 s can still reach 0.241 mA/cm^2^, about 5.7 times that of the commercial catalyst. It indicated that the catalytic oxidation stability of the high index facets catalyst synthesized under this condition was the best.

#### 3.2.4. Anti-CO Poisoning Test of the Catalysts

In order to further test the anti-CO of the samples, the test solution was 0.5 mol/L H_2_SO_4_ solution. During the test, CO gas was continuously injected, and the sweep rate was 50 mV/s. The test results are shown in Figure 8. In the test, the lower the CO oxidation peak potential of the catalyst is, the stronger the anti-CO toxicity of the catalyst is. It can be found from the test results that the CO oxidation peak potentials of the synthesized high-index Pt-based catalysts were 0.68 V for PVP:Pt = 50:1, 0.70 V for PVP:Pt = 75:1, 0.725 V for PVP:Pt = 25:1, 0.73 V for PVP:Pt = 100:1 and 0.735 V for PVP:Pt = 0:1.

This is because when PVP is not added, the formed nanocrystals will agglomerate to form some large particles. The active sites cannot be fully exposed, resulting in poor anti-toxicity and reduced oxidation current density. The dispersion of catalyst grains increased with the increase of PVP content. When PVP:Pt = 50:1, the CO oxidation peak potential of the synthesized catalyst was 0.68 V, which was the lowest compared with other catalysts, indicating that it had the best anti-CO toxicity.

#### 3.2.5. Variable-Temperature Cyclic Voltammetry of the Catalysts

In the cause of investigate the effect of catalyst on the catalytic activity of ethanol in the range of 20 °C~60 °C, the temperature-dependent cyclic voltammetry was conducted. The test results are shown in Figure 9. At the same time, the measured temperature curve is fitted, and the measured current density is calculated with the Arrhenius equation:i_p_ = kexp[−(W/R) × (1/T)](6)

In Formula (6), i_p_ represents peak current density, R represents gas constant 8.314 J/(mol·K), K represents Boltzmann constant, W represents Arrhenius activation energy and T represents temperature.
lni_p_ = [−(W/R) × (1/T)] + lnk(7)
k = −W/R(8)

Figure 9 is made with 1/T as abscissa and lni_p_ as ordinate using Origin for variable temperature fitting. After fitting, the fitting slope of each group of catalysts at variable temperature can be obtained by fitting the graph. Then, the activation energy W of each group of catalysts for catalytic oxidation of alcohols can be calculated by Formula (8), and the activation energy shown in Table 3 can be obtained. It can be known from Reference that the activation energy of the catalyst is closely related to the catalytic performance of the catalyst [32]. The lower the activation energy is, the easier the catalyst starts to participate in the reaction, and the higher the catalytic performance of the catalyst is. From the Table 3, the activation energies of the prepared catalysts are higher than those of commercial Pt/C catalysts. It was found that the activation energies of the high index facets catalysts prepared with different PVP additions were also different. Only when PVP:Pt = 50:1, the activation energy of the catalyst was the poorest and the catalytic performance was the best, which was consistent with the previous electrochemical analysis.

#### 3.2.6. Cyclic Voltammetry Analysis of Degradation of the Catalysts

In order to measure the cyclic sustainability of the catalyst, the stability of the catalyst was tested for 500 cycles. Figure 10 is the cyclic voltametric failure diagram of the catalyst for ethanol. It can be seen from the figure that with the add in the number of scanning cycles, the peak current density of the catalysts decreases. This is because in the process of catalytic oxidation of C_2_H_5_OH, many intermediate products without complete oxidation will be produced. These intermediate products will be adsorbed on the active site of the Pt catalyst, which prevents the further oxidation of ethanol from causing the decrease of the oxidation current density. In the sustainability test of the catalyst, the oxidation current density of ethanol decreases with the addition of the number of cycles. 

However, in the test process, the degree of exhaustion at each stage was different. At the final 500 cycles, the retention rate of the catalyst on the oxidation current density of C_2_H_5_OH was also different. By taking the oxidation current density per 100 cycles and the oxidation current density per 1 cycle, the calculation results are shown in Table 4. It can be seen from the table that the retention rates of catalysts in each group were 60.00%, 65.88%, 70.87%, 84.14%, 72.23% and 70.80%, separately. It can be seen that the retention rate of ethanol oxidation current density of the prepared high index crystal surface catalyst is better than that of commercial Pt/C catalyst. When the addition amount of PVP was PVP:Pt= 50:1, the stability of the prepared high index facets catalyst was the best, indicating that the catalyst at this time exposed sufficient active sites, had the best dispersion and the highest stability for ethanol oxidation.

## 4. Results and Discussion of Catalysts with Glycine as Variable

### 4.1. Characterization

#### 4.1.1. X-ray Diffraction (XRD) Characterization of the Catalysts

XRD patterns of catalysts with different glycine additions, as shown in Figure 11. It can be found from the figure that the prepared catalysts all had five peaks in the range of 10°–90°. Compared with the Pt standard PDF card, it was found that the peaks at 39.7° corresponded to the Pt (111) crystal plane, 46.2° corresponded to the Pt (200) crystal plane, 67.5° corresponded to the Pt (220) crystal plane, 81.3° corresponded to the Pt (311) crystal plane, 85.7° corresponded to the Pt (222) crystal plane, indicating that the prepared catalyst Pt^4+^ was successfully reduced to Pt.

#### 4.1.2. SEM Characterization of the Catalysts

Glycine as a morphology regulator plays a very important role in crystal morphology control synthesis. It can control the growth rate of Pt-based nanocrystals by changing the amount of glycine to adjust the morphology of Pt-based nanoparticles. Therefore, under the same conditions, five groups of different catalyst control groups were prepared by changing the addition quantity of glycine to explore the influence of the addition amount of glycine on the morphology of Pt HIFs. The molar ratios of glycine to Pt were Gly:Pt = 0:1, Gly:Pt = 8:1, Gly:Pt = 16:1, Gly:Pt = 24:1 and Gly:Pt = 32:1, respectively. The catalysts were characterized with SEM, as shown in Figure 12.

From Figure 12a, it can be seen that when glycine is not added, the catalyst is some smaller spherical particles. When the addition amount of glycine was Gly:Pt = 8:1, it was found that the formed catalyst nanoparticles were mainly some concave cubes and some irregular polyhedrons, and the size of nanoparticles increased compared with that without glycine, indicating that PVP was reductive. When the addition of glycine was Gly:Pt = 16:1, it was found that the catalyst particles were mainly concave cubic particles with stable morphology. When the addition of glycine was Gly:Pt = 24:1, it was found that the morphology of Pt catalyst nanoparticles could be clearly observed. Compared with other different glycine addition ratios, the size of nanoparticles increased. When the addition of glycine is Gly:Pt = 32:1, it can be seen that the catalyst nanoparticles are basically stable without other irregular polyhedrons, but the nanoparticles have greatly increased, which may generate the formation of other various high index crystal facets. These effects show that the addition of suitable glycine can regulate the reduction kinetics of Pt concave nanocubes, and then control the morphology of nanoparticles, indicating that glycine plays an essential role in contour and size control in this process [33,34].

#### 4.1.3. TEM Characterization of the Catalysts

For the benefit of investigate the effect of glycine on the formation of Pt high index facets, the catalyst was characterized with HTEM. By means of Figure 13, it can be seen that the morphology of Pt high index facets catalysts change obviously with the addition of glycine. The larger the concave angle of catalyst nanoparticles and the higher the step atomic density and the exposed active sites, the higher the catalytic performance [22]. It can be seen in Figure 13a that when glycine was not added in the preparation process, it was found that the prepared catalyst only formed some small nanoparticles, and the nanoparticles aggregated to form clusters, and the particle size was about 12.50 nm. When Gly:Pt = 8:1, it was found that the prepared nanoparticles formed close to the cube, and the particle size was 18.12 nm. The exposed crystal plane was mainly {100} crystal plane (Table 5). When Gly:Pt = 16:1, it was found that the synthesized catalyst nanoparticles began to concave inward from the square morphology, and the statistical particle size was 21.52 nm. It was found that the exposed high index crystal faces were mainly {410} and {720}. When Gly:Pt = 24:1, the prepared catalyst nanoparticles can clearly see a stable concave cube with a particle size of 26.72 nm. By measuring the concave angle, it is found that the exposed high-index crystal planes are mainly {310}, {520} and {830}. When Gly:Pt = 32:1, the morphology of the catalyst nanoparticles was stable, and the particle size was 28.24 nm. It was found that the exposed high-index facets were mainly {310}, {720} and {830} by measuring the concave angle. It was found that with the addition of glycine, the morphology of the nanocatalyst gradually increased from the initial amorphous cluster to the complete formation of concave cubes. When Gly:Pt = 32:1, the morphology of the nanoparticles is basically stable, but the nanoparticles are larger because of excessive glycine to accelerate the growth of the nanoparticles. It was found with observation that with the increase of glycine, the catalyst nanoparticles gradually changed from spherical to cubic and finally stabilized to concave cubic morphology. Therefore, glycine played a crucial role in the synthesis of high index facets crystalline nanocatalysts [35].

### 4.2. Electrochemical Characterization

#### 4.2.1. Electrochemical Active Surface Area of Catalyst

The H-adsorption–desorption curve of the prepared sample with commercial Pt/C is shown in Figure 14. It can be seen that the oxidation peak appears in the range of −0.3 V~−0.1 V during the positive scanning process, and the reduction peak appears in the range of −0.2 V~−0.3 V during the negative scanning process. According to Formulas (1) and (2), the electrochemical active area (ESA) in Table 6 can be obtained with a calculation. It can be found that compared with commercial Pt/C catalysts, the electrochemical active area of the prepared high-index facets catalysts has been greatly improved. With the increase of glycine content, the electrochemical active area first increased and then decreased. When glycine molar ratio was Gly:Pt = 24:1, the electrochemical activity area reached the maximum of 2.194 m^2^/g. The analysis results showed that glycine acted as a reducing agent and a surface morphology control agent in the formation of high-index facets. When the amount of glycine is small, the catalyst is square and irregular polyhedron, and the electrochemical active area is low. This is because the specific surface area of the cube is small and the active site is less than that of the concave cube or convex cube. When the amount of glycine added was Gly:Pt = 24:1, it reached the maximum. When the addition amount of glycine continued to increase, it could be found that the electrochemical active area began to decrease again. The reason may be that with the increase of glycine content, massive other irregular polyhedrons are formed, and the grain size becomes larger, the specific surface area decreases, and the electrochemical active area decreases.

#### 4.2.2. Cyclic Voltammetry of the Catalysts in Ethanol Oxidation

The catalytic oxidation of ethanol over catalysts prepared with different amounts of glycine is shown in Figure 15a. It can be found that the peak current density of the prepared high index facets catalyst has been greatly improved compared with that of the commercial Pt/C catalyst. This is because there are a large number of step atoms and twist atoms on the surface of the high index facets, which can be used as the active sites for catalytic ethanol. It is easier to contact with ethanol so that the chemical bond of ethanol is exercised and becomes the center of the chemical reaction. Figure 15b shows the corresponding mass ratio active column diagram. It can be seen in Figure 15a that with the addition of glycine content, the oxidation current density of catalyst for ethanol first increased and then decreased. When glycine was not added to the reaction system, it was found that the oxidation current density of ethanol was only 0.906 mA/cm^2^. With the increase of glycine content, the maximum value of 2.052 mA/cm^2^, which was about 5.8 times that of commercial Pt/C catalyst when Gly:Pt = 24:1. The catalyst synthesized under this condition has the best catalytic activity for ethanol. Further increasing the addition of glycine, it was found that the oxidation current density of ethanol began to decrease. This is because as a reducing agent and surface morphology control agent, excessive glycine will make the catalyst grain continue to grow and the oxidation current density decrease.

#### 4.2.3. I–t Curves of the Catalysts

The chronoamperometry of each set of catalysts is shown in Figure 16. It can be found from the figure that the current density gradually decreases with the increase of time, and tends to be stable at about 1000 s. This is because at the beginning, the ethanol concentration near the electrode is high and the diffusion layer is thin, resulting in a high current density. However, with the extension of time, the concentration of ethanol molecules near the electrode is declining.

At the same time, the incomplete oxidation products of ethanol will be adsorbed on the surface of the active site of the Pt catalyst, resulting in that the catalyst cannot continuously catalyze the oxidation of ethanol, and the current density is getting lower and lower. Therefore, the timing current test of the catalyst can measure the catalytic performance and anti-poisoning performance of the catalyst. As shown in Table 6, when Gly:Pt = 24:1, the oxidation current density of the catalyst can still reach 0.241 mA/cm^2^ after 1100 s, which is about 5.7 times of the commercial catalyst. It shows that the high index crystal surface catalyst synthesized under this condition has the most optimum catalytic oxidation stability for C_2_H_5_OH.

#### 4.2.4. Anti-CO Poisoning Test of the Catalysts

The anti-CO toxicity tests of the catalysts are shown in Figure 17. The lower the oxidation peak potential of CO on the catalyst, the stronger the resistance of the catalyst to CO poisoning. It can be seen in the test results that the CO oxidation peak potentials of the synthesized high index facets Pt-based catalysts were 0.68 V(Gly:Pt = 24:1), 0.715 V(Gly:Pt = 32:1), 0.72 V(Gly:Pt = 16:1), 0.73 V(Gly:Pt = 8:1) and 0.75 V(Gly:Pt = 0:1), respectively. Glycine plays a role of reducing agent and morphology control in the synthesis of catalyst. When Gly:Pt = 24:1, the synthesized catalyst shows the best concave cube, and the measured peak potential of CO oxidation is 0.68 V, which shows the best anti-CO poisoning performance [36,37,38].

#### 4.2.5. Variable-Temperature Cyclic Voltammetry of the Catalysts

The cv curves of each group were fitted by Origin as shown in Figure 18. The fitting slope of each group of catalysts at variable temperature can be obtained by fitting the figure, and then the activation energy W of each group of catalysts for catalytic oxidation of alcohols can be calculated with Formula 8, and the activation energy data shown in Table 7 can be obtained. It can be known from Reference that the reactivation energy of the catalyst is compactly related to the catalytic performance of the catalyst [32]. The lower the activation energy is, the easier the catalyst starts to take part in the reaction, and the higher the catalytic performance of the catalyst is.

It can be seen from Table 7 that the activation energy of the prepared catalysts is higher than that of commercial Pt/C (JM) catalysts. Moreover, it can be found that the activation energies of the high index facets catalysts prepared with different glycine additions are also different. Only when Gly:Pt = 24:1, the activation energy of the catalyst is the lowest of 49.57 kJ/mol, showing the optimal catalytic performance, which is consistent with the previous electrochemical analysis.

#### 4.2.6. Cyclic Voltammetry Analysis of Degradation of the Catalysts

Figure 19 is the cyclic voltammetry curve of the catalytic oxidation current density of C_2_H_5_OH with the number of scan cycles. It can be got from the figure that with the increase of the number of scanning cycles, the peak current density of the catalysts declined. This is because in the process of catalytic oxidation of C_2_H_5_OH, many intermediate products without complete oxidation will be produced. These intermediate products will be adsorbed on the active site of the Pt catalysts, which prevents the further oxidation of ethanol from causing the decrease of the oxidation current density. In the sustainability test of the catalyst, the oxidation current density of ethanol decreases with the increase of the number of cycles. However, in the test process, the degree of exhaustion at each stage is different. At the final 500 cycles, the retention rate of the catalyst on the oxidation current density of ethanol is also different. By taking the oxidation current density at each 100 cycles and the oxidation current density at the first cycle, the calculation results are shown in Table 8. The retention rates of the catalysts were 60.00%, 65.88%, 70.87%, 84.14%, 73.23% and 70.80%, respectively. It can be concluded that the retention rate of the high index crystal surface catalyst for ethanol oxidation is higher than that of commercial Pt/C catalyst. When the addition amount of glycine was Gly:Pt = 24:1, the stability of the prepared high-index crystal appearance catalyst was the best, indicating that the catalyst at this time exposed sufficient active sites, had the best dispersion, and had the highest stability for ethanol oxidation. Glycine as appearance morphology regulation agent and auxiliary reducing agent plays an essential part in the preparation of high index crystal facets. When the glycine content was low, the formed nanoparticles were mainly composed of some cubes and irregular polyhedrons, and the active sites were not fully exposed. When the glycine content increased, the reduction rate increased with the increase of the amount of reducing agent, resulting in the excessive growth of nanoparticles, and the active sites began to disappear, leading the decrease of ethanol oxidation ability.

## 5. Conclusions

Pt high-index crystal surface nanocatalyst was prepared with the hydrothermal method and in-situ growth method. By analyzing the crystal faces, it is found that the exposed crystal faces of the Pt high index facets nanocatalysts are mainly {310}, {520} and {830}. This enables the controllable synthesis of highly indexed facet oriented nanocrystals with excellent electrocatalytic oxidation performance for ethanol.

By adjusting the addition amount of PVP in the shaping process of high index facets, it can be found that PVP plays the role of dispersant and reducing agent in the formation process of Pt high index facets. When the mass ratio is PVP:Pt = 50:1, uniform high index facets crystalline nanoparticles are formed. Its electrochemical active area is 2.194 m^2^/g, the peak current density is 2.052 mA/cm^2^, and the stability is 0.241 mA/cm^2^. After 500 cycles of failure test, the current density retention rate is the highest.

By adjusting the addition amount of glycine in the shaping process of high index facets, it can be found that glycine plays the role of morphology control and auxiliary reduction in the shaping process of Pt high index facets catalyst. When the molar ratio is Gly:Pt = 24:1, uniform high index facets crystalline nanoparticles are formed. The electrochemical active area, peak current density and stability are the highest. After 500 cycles of failure test, the current density retention rate is the highest.

## Figures and Tables

**Figure 1 nanomaterials-12-04451-f001:**
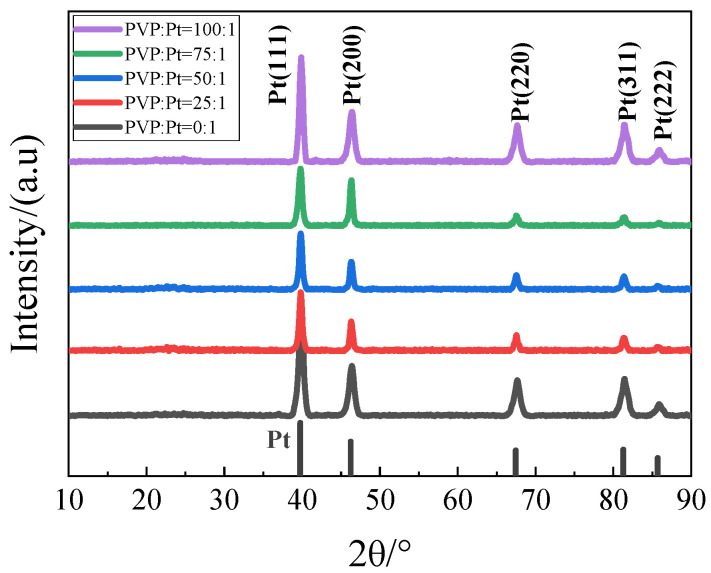
XRD patterns of catalysts in each group.

**Figure 2 nanomaterials-12-04451-f002:**
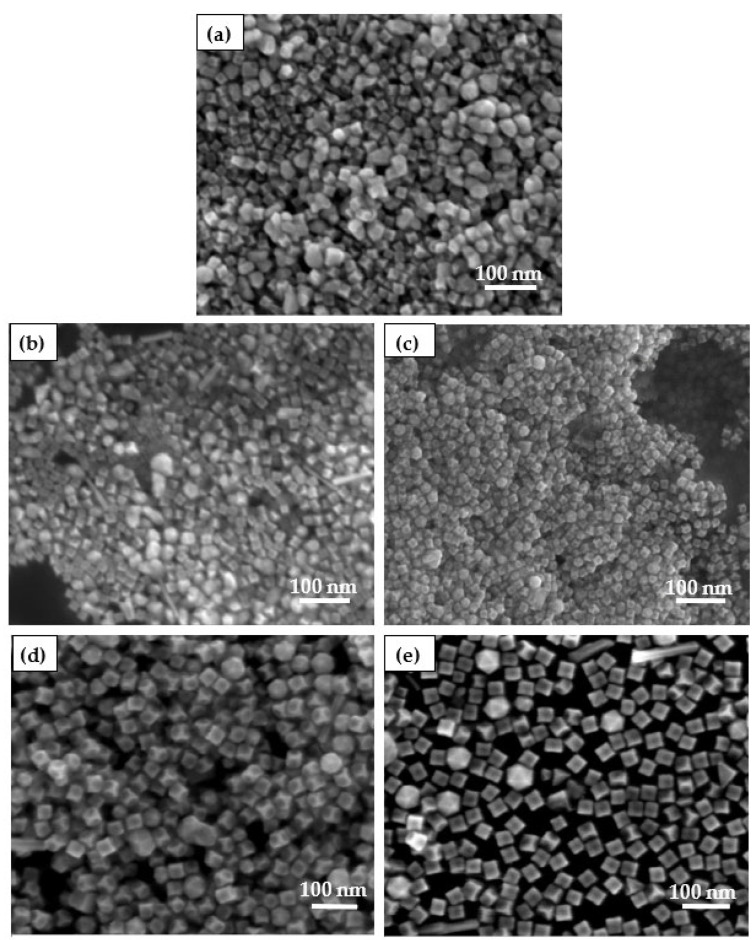
Catalysts prepared by different PVP: (**a**) PVP:Pt = 0:1; (**b**) PVP:Pt = 25:1; (**c**) PVP:Pt = 50:1; (**d**) PVP:Pt = 75:1; (**e**) PVP:Pt = 100:1.

**Figure 3 nanomaterials-12-04451-f003:**
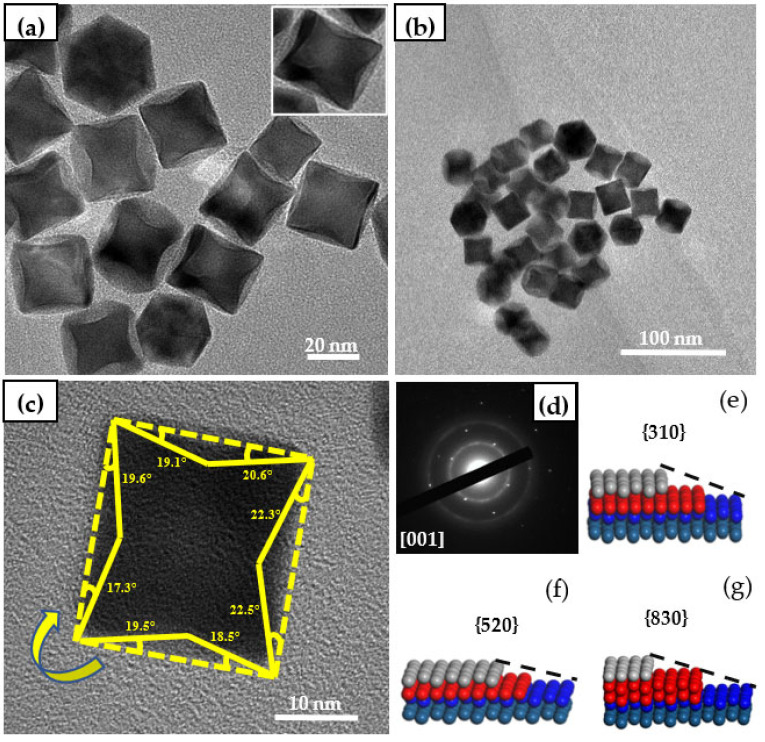
TEM images of Pt high-index nanocatalysts: (**a**,**b**) are TEM images, (**c**) are TEM images of concave angle measurement; (**d**) Electron diffraction spots from direction; (**e**–**g**) are three typical high index crystal face models.

**Figure 4 nanomaterials-12-04451-f004:**
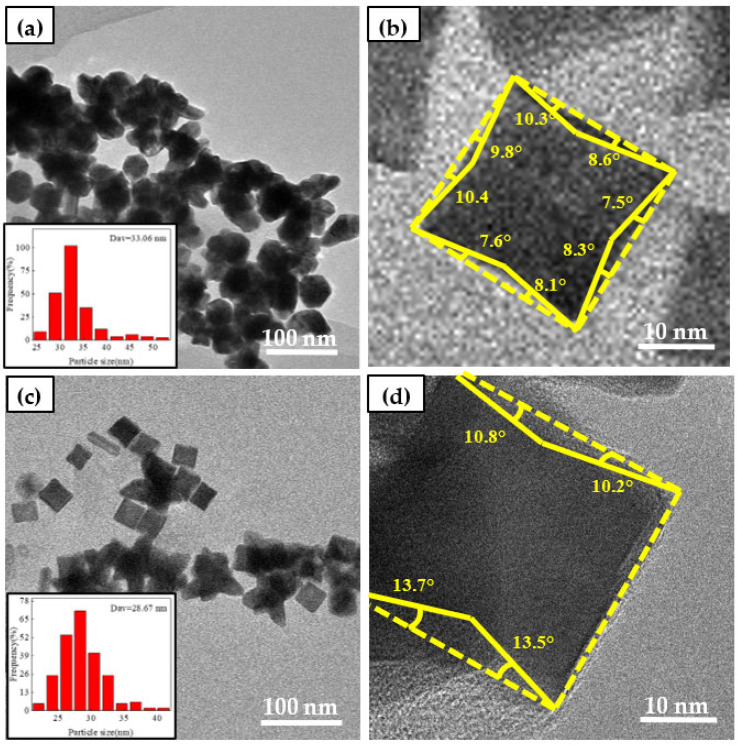
Catalysts prepared by different PVP:(**a**,**b**); PVP:Pt = 0:1; (**c**,**d**) PVP:Pt = 25:1; (**e**,**f**) PVP:Pt = 50:1; (**g**,**h**) PVP:Pt = 75:1; (**i**,**j**)PVP:Pt = 100:1.

**Figure 5 nanomaterials-12-04451-f005:**
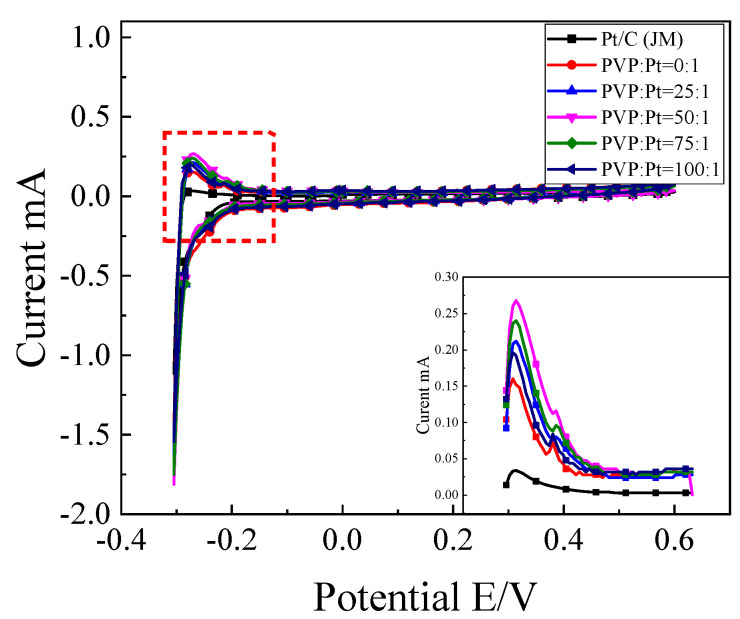
CV curve of catalyst in 0.5 mol/L H_2_SO_4_ solution.

**Figure 6 nanomaterials-12-04451-f006:**
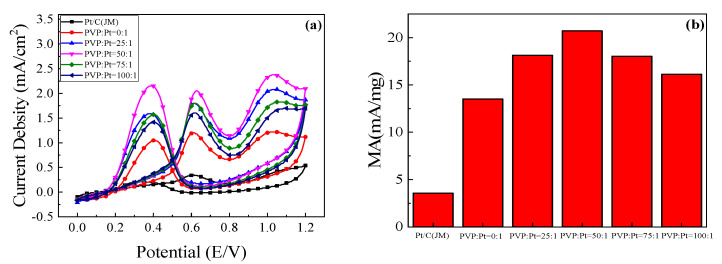
Catalyst in 0.5 mol/L H_2_SO_4_+1 mol/L C_2_H_5_OH solution: (**a**) Cyclic voltammetry curve; (**b**) Column chart of mass ratio activity.

**Figure 7 nanomaterials-12-04451-f007:**
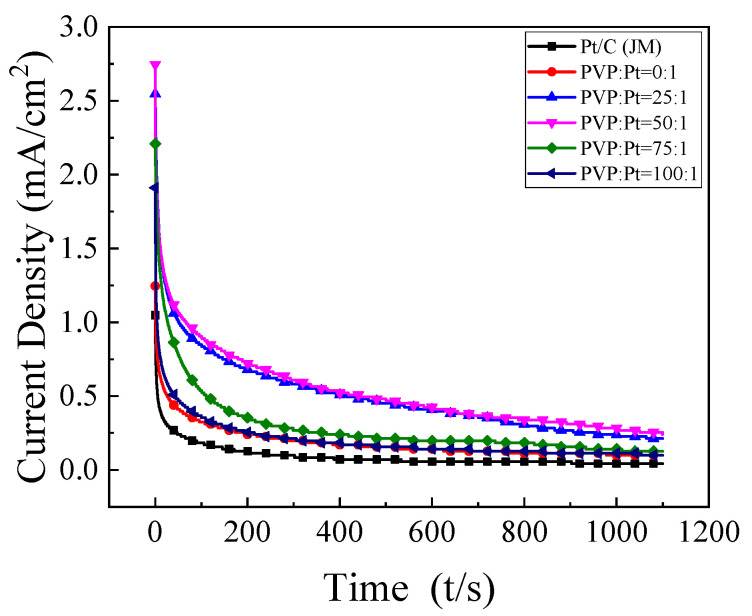
Steady-state current density curve of samples in 0.5 mol/L H_2_SO_4_+1 mol/L C_2_H_5_OH solution.

**Figure 8 nanomaterials-12-04451-f008:**
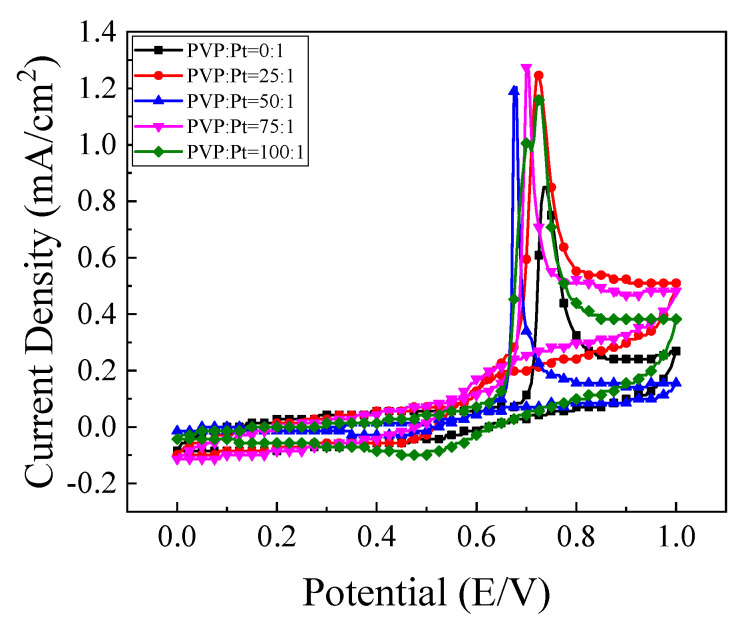
Anti-CO poisoning curve of catalyst in 0.5 mol/L H_2_SO_4_ solution.

**Figure 9 nanomaterials-12-04451-f009:**
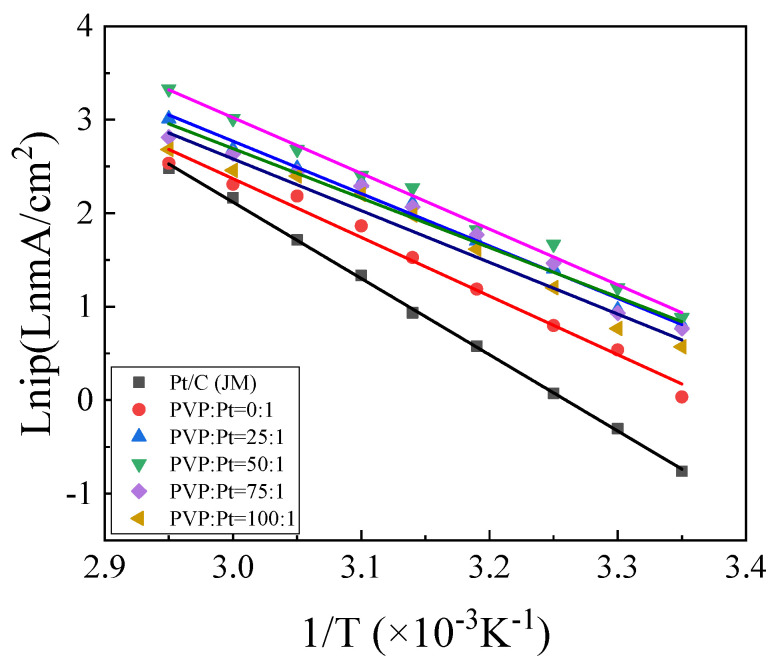
Allenius curve.

**Figure 10 nanomaterials-12-04451-f010:**
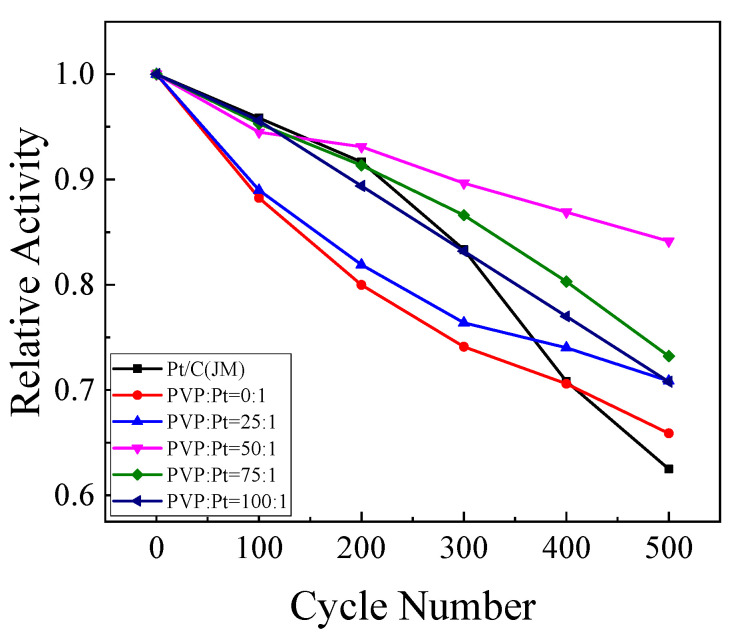
The peak current density of ethanol oxidation decreased with the number of scan cycles.

**Figure 11 nanomaterials-12-04451-f011:**
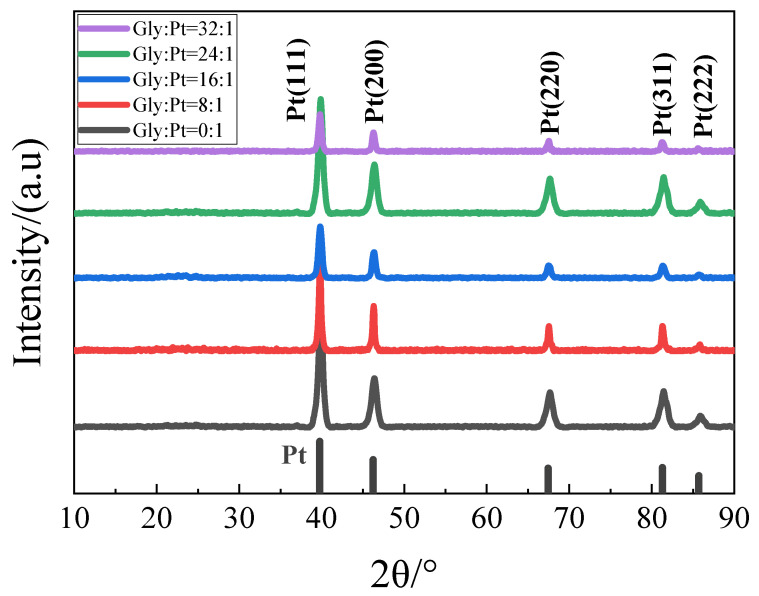
XRD patterns of catalysts in each group.

**Figure 12 nanomaterials-12-04451-f012:**
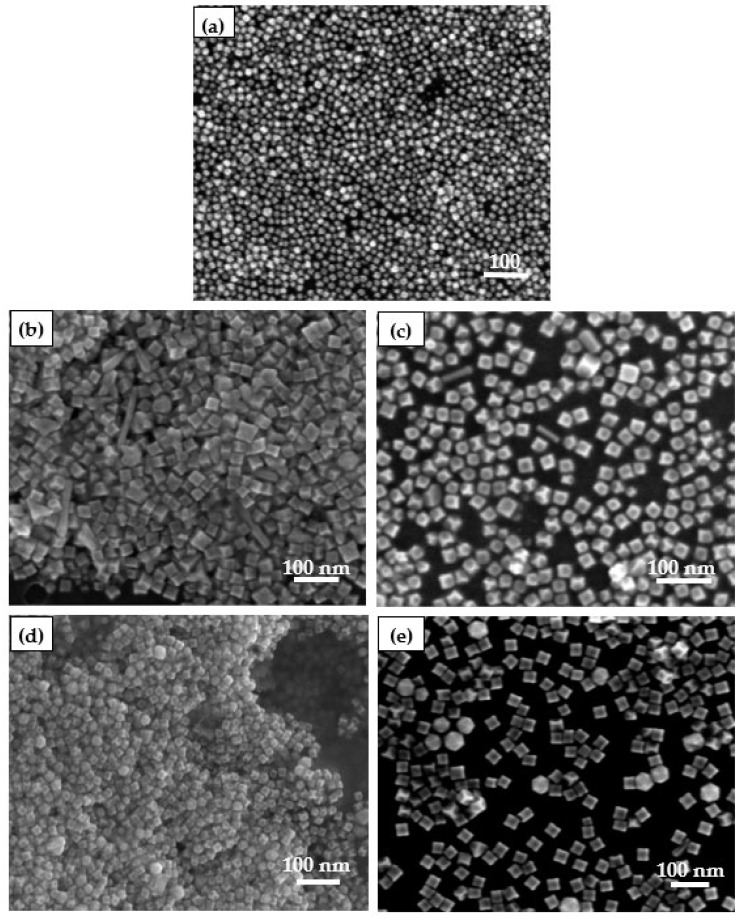
Catalysts prepared by different glycine:(**a**) Gly:Pt = 0:1; (**b**) Gly:Pt = 8:1; (**c**) Gly:Pt = 16:1; (**d**) Gly:Pt = 24:1; (**e**) Gly:Pt = 32:1.

**Figure 13 nanomaterials-12-04451-f013:**
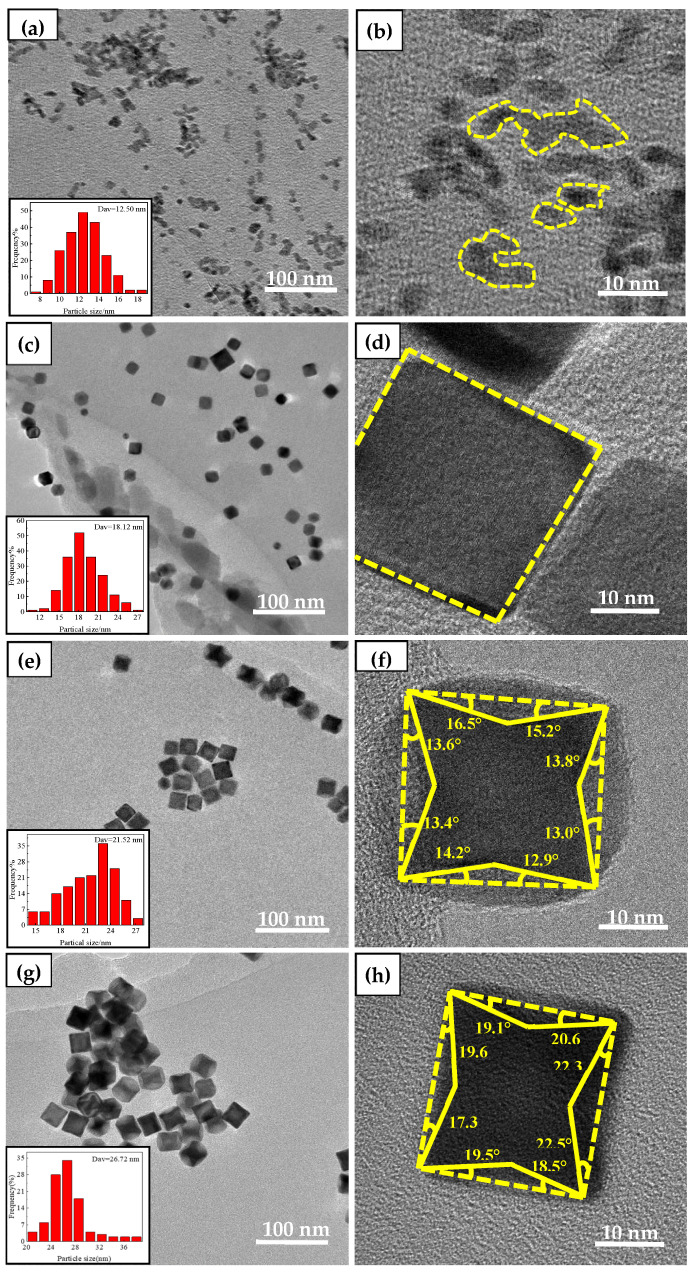
Catalysts prepared by different glycine: (**a**,**b**) Gly:Pt = 0:1; (**c**,**d**) Gly:Pt = 8:1; (**e**,**f**) Gly:Pt = 16:1; (**g**,**h**) Gly:Pt = 24:1; (**i**,**j**) Gly:Pt = 32:1.

**Figure 14 nanomaterials-12-04451-f014:**
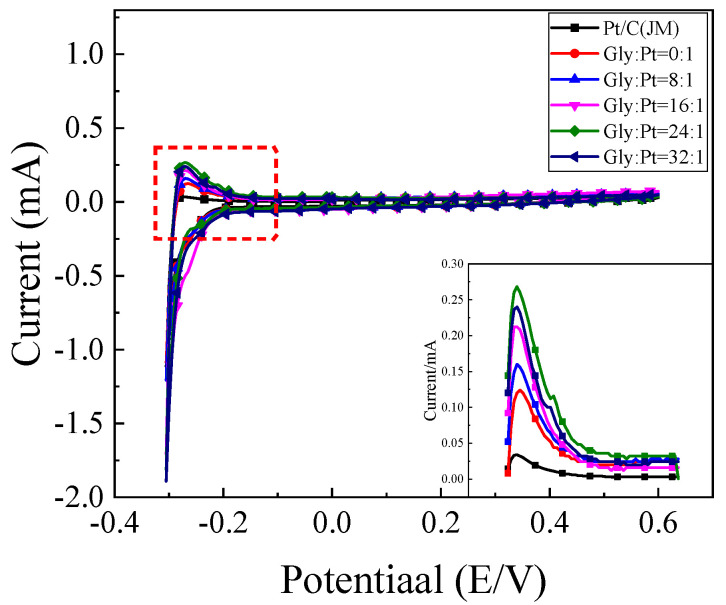
H adsorption-desorption curve of the catalyst in 0.5 mol/L H_2_SO_4_ solution.

**Figure 15 nanomaterials-12-04451-f015:**
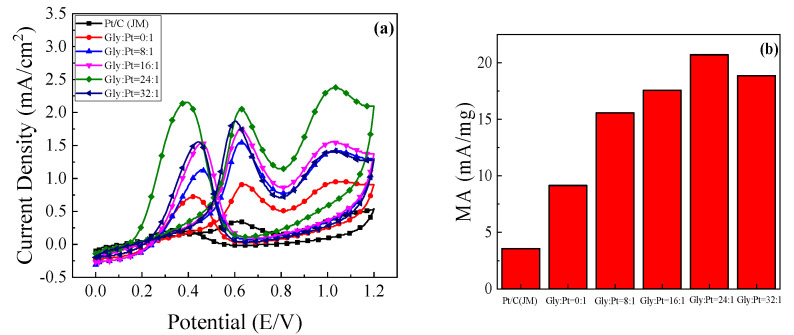
The (**a**) cyclic voltammetry curve of the catalyst in0.5 mol/L H_2_SO_4_+1 mol/L C_2_H_5_OH solution; (**b**) Column chart of mass ratio activity.

**Figure 16 nanomaterials-12-04451-f016:**
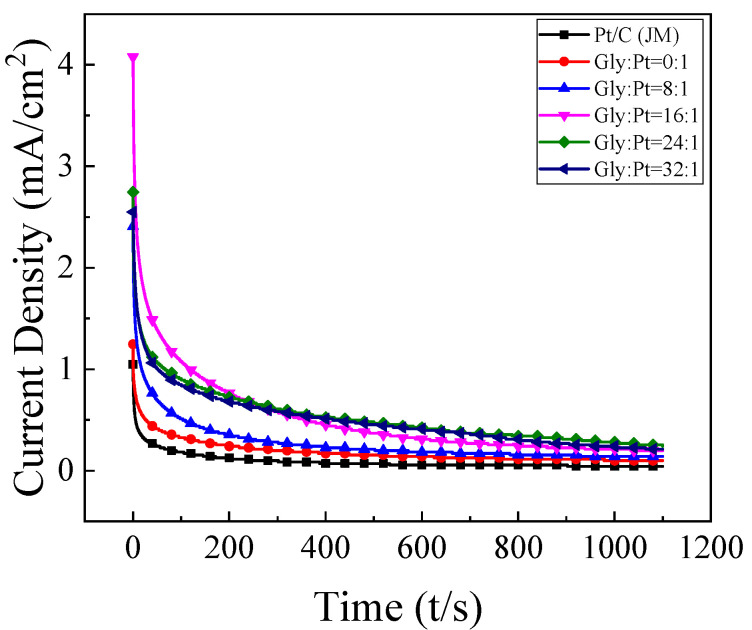
Steady-state current density curve of catalyst in 0.5 mol/L H_2_SO_4_+1 mol/L C_2_H_5_OH solution.

**Figure 17 nanomaterials-12-04451-f017:**
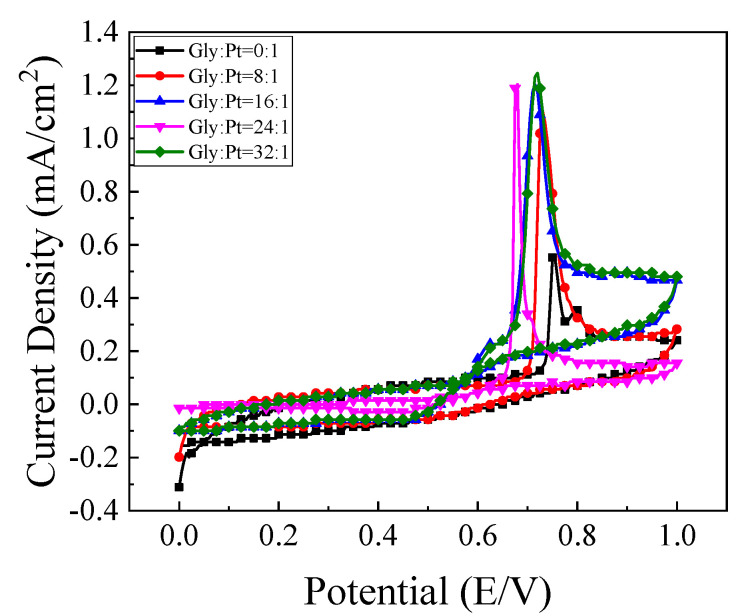
Anti-CO poisoning curve of catalyst in 0.5 mol/L H_2_SO_4_ solution.

**Figure 18 nanomaterials-12-04451-f018:**
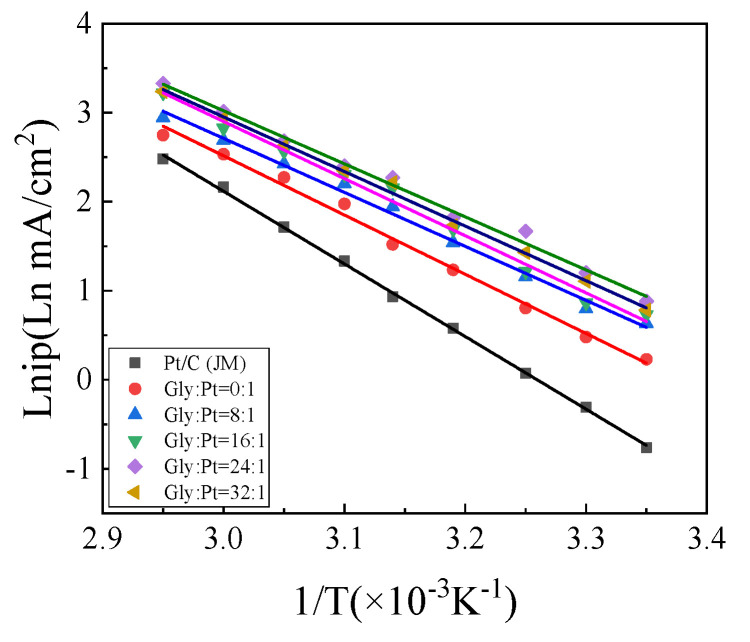
Allenius curve.

**Figure 19 nanomaterials-12-04451-f019:**
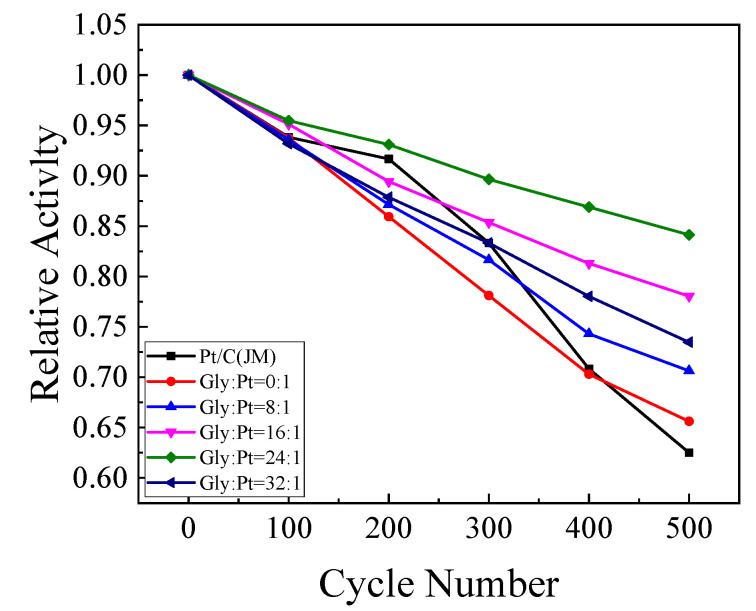
Decrease in current density of ethanol oxidation over catalysts with scanning cycles.

**Table 1 nanomaterials-12-04451-t001:** Exposed crystal planes of nanocatalysts with high index facets prepared by different PVP additions.

Sample	Angle/°	Crystal Facets
PVP:Pt = 0:1	8.1, 9.5	{710};{610}
PVP:Pt = 25:1	9.5, 11.3, 14.0	{610};{510};{410}
PVP:Pt = 50:1	18.4, 20.6, 21.8	{310};{830};{520}
PVP:Pt = 75:1	14.0, 15.9	{410};{720}
PVP:Pt = 100:1	14.0, 15.9	{410};{720}

**Table 2 nanomaterials-12-04451-t002:** Electrochemical active surface area, oxidation current density and steady state current density of samples.

Sample	ESA (m^2^/g)	Peak Current Density (mA/cm^2^)	Steady Current Density (mA/cm^2^)
Pt/C(JM)	0.239	0.354	0.042
PVP:Pt = 0:1	1.203	1.340	0.098
PVP:Pt = 25:1	1.651	1.797	0.212
PVP:Pt = 50:1	2.194	2.052	0.241
PVP:Pt = 75:1	1.911	1.786	0.127
PVP:Pt = 100:1	1.641	1.599	0.099

**Table 3 nanomaterials-12-04451-t003:** Fitting Analysis Data of Catalyst Temperature Variation.

Sample	Slope k	Activation Energy W (kJ/mol)
Pt/C (JM)	−8.16	67.84
PVP:Pt = 0:1	−6.62	55.04
PVP:Pt = 25:1	−6.35	52.79
PVP:Pt = 50:1	−5.96	49.57
PVP:Pt = 75:1	−6.16	51.21
PVP:Pt = 100:1	−6.48	52.87

**Table 4 nanomaterials-12-04451-t004:** Current density retention rate of catalyst.

Sample	1-Cycle and 500-Cycle Peak Current Density (mA/cm^2^)	Current Density Retention Rate/%
Pt/C (JM)	0.354/0.221	62.50
PVP:Pt = 0:1	1.340/0.883	65.88
PVP:Pt = 25:1	1.797/1.417	70.87
PVP:Pt = 50:1	2.052/1.727	84.14
PVP:Pt = 75:1	1.786/1.308	73.23
PVP:Pt = 100:1	1.599/1.132	70.80

**Table 5 nanomaterials-12-04451-t005:** Preparation of high-index nanocatalysts with different glycine content exposed crystal face.

Sample	Angle/°	Crystal Face
Gly:Pt = 0:1	---	---
Gly:Pt = 8:1	0	{100}
Gly:Pt = 16:1	14.0, 15.9	{410};{720}
Gly:Pt = 24:1	18.4, 20.6, 21.8	{310};{830};{520}
Gly:Pt = 32:1	15.9, 18.4, 21.8	{720};{310};{830}

**Table 6 nanomaterials-12-04451-t006:** Electrochemical active surface area, oxidation current density and steady state current density of catalysts.

Sample	ESA (m^2^/g)	Peak Current Density (mA/cm^2^)	Steady Current Density (mA/cm^2^)
Pt/C(JM)	0.239	0.354	0.042
Gly:Pt = 0:1	1.086	0.906	0.099
Gly:Pt = 8:1	1.352	1.543	0.142
Gly:Pt = 16:1	1.462	1.741	0.198
Gly:Pt = 24:1	2.194	2.052	0.241
Gly:Pt = 32:1	1.761	1.868	0.212

**Table 7 nanomaterials-12-04451-t007:** Fitting Analysis Data of Catalyst Temperature Variation.

Sample	Slope k	Activation Energy W (kJ/mol)
Pt/C(JM)	−8.16	67.86
Gly:Pt = 0:1	−6.66	55.34
Gly:Pt = 8:1	−6.57	54.69
Gly:Pt = 16:1	−6.42	53.39
Gly:Pt = 24:1	−5.96	49.57
Gly:Pt = 32:1	−6.13	50.98

**Table 8 nanomaterials-12-04451-t008:** Current density retention of catalyst.

Sample	1-Cycle and 500-Cycle Peak Current Density (mA/cm^2^)	Current Density Retention rate/%
Pt/C(JM)	0.354/0.221	62.50
Gly:Pt = 0:1	0.906/0.595	65.63
Gly:Pt = 8:1	1.543/1.090	70.64
Gly:Pt = 16:1	1.741/1.359	78.05
Gly:Pt = 24:1	2.052/1.727	84.14
Gly:Pt = 32:1	1.868/1.373	73.48

## Data Availability

Not applicable.

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
