# Peer review of "One-Pot Synthesis of Pt High Index Facets Catalysts for Electrocatalytic Oxidation of Ethanol"

_nanomaterials, 2022, doi:10.3390/nano12244451_

Round 1
Reviewer 1 Report
The paper describes properties of Pt high-index facets nanocatalyst for direct ethanol fuel cell. At present, the DEFC electrocatalyst mainly uses Pt as the main active component, and other transition metals and oxides are added as the auxiliary catalysts. Pt high-index facets nanocatalyst was prepared by one-pot hydrothermal method by adjusting the amount of PVP and glycine. When the mass ratio of PVP and Pt was 50:1 and the molar ratio of glycine and Pt was 24:1, Pt nanocatalysts with {310}, {520} and {830} high exponential facets were prepared. XRD, TEM and SEM were used to characterize the micro-structure of the nanocatalyst, and the influence of PVP and glycine on the synthesis of high-index facets catalyst was studied. Authors have been found that the performance of the prepared catalyst was better than that of commercial catalyst.
The manuscript is written clearly and understandably without frills. All conclusions supported by the results.
Sometimes abbreviations are given in the text without decoding. For example, PVP (line 18), XRD, TEM (line 20), SEM (line 21). It is necessary to provide transcripts of abbreviations at their first appearance in the text.
The paper can be published in Nanomaterials after minor revision.
Author Response
Thank you for your good suggestions. We have changed and marked in the manuscript.

Reviewer 2 Report
The article "One-Pot synthesis of Pt High Index facets Catalysts for Electrocatalytic Oxidation of Ethanol" written by Guo et al. describe preparation of suitable Pt catalyst for ethanol oxidation. Electrochemically, they show pretty good stability of the catalyst after 500 cycles. Could they also provide some electron microscopy images and XRD data for catalyst after the 500 cycles?
The article is readily readable and provide clear story how to design and test the Pt catalyst for ethanol oxidation. I will recommend it for publication.
Author Response
After 500 cycles of Stability tests, due to the nano-catalyst tightly attached to the electrode surface and the amount is very small, how to carry out further micro-test on the premise of ensuring that it is not contaminated is currently in the exploration, our future research work will be solved in time, very grateful for understanding.
